# Drought Monitoring over Nepal for the Last Four Decades and Its Connection with Southern Oscillation Index

Damodar Bagale *, Madan Sigdel  and Deepak Aryal

Central Department of Hydrology and Meteorology, Tribhuvan University, Kathmandu 44613, Nepal;
sigdelbro@gmail.com (M.S.); deepak.aryal@cdhm.tu.edu.np (D.A.)
* Correspondence: damu.bagale@gmail.com

**Abstract:** This study identified summer and annual drought events using the Standard Precipitation Index (SPI) for 107 stations across Nepal from 1977 to 2018. For this, frequency, duration, and severity of drought events were investigated. The SPI4 and SPI12 time scales were interpolated to illustrate the spatial patterns of major drought episodes and their severity. A total of 13 and 24 percent of stations over the country showed a significant decreasing trend for SPI4 and SPI12. Droughts were recorded during El Niño and non-El Niño years in Nepal. Among them, 1992 was the worst drought year, followed by the drought year, 2015. More than 44 percent of the locations in the country were occupied under drought conditions during these extreme drought events. Droughts have been recorded more frequently in Nepal since 2005. The areas of Nepal affected by extreme, severe, and moderate drought in summer were 8, 9, and 18 percent, while during annual events they were 7, 11, and 17 percent, respectively. Generally, during the drought years, the SPI and Southern Oscillation Index (SOI) have a strong phase relation compared to the average years.

**Keywords:** drought; El Niño; Mann-Kendall test; Nepal; SOI; SPI



## 1. Introduction

Drought is a typical natural locally or regionally recurring characteristic of the climate. It occurs in virtually all climatic regimes in areas with high and low rainfall. A universal definition of drought is an unrealistic expectation [1]. Drought is a temporary aberration, in contrast to aridity, a permanent feature of the climate, and is restricted to low rainfall areas [1]. Drought is considered by many to be the most complex but least understood of all-natural hazards, affecting more people and the environment, causing more substantial economic losses than any other hazard [2,3]. The effects of drought often accumulate slowly over a considerable time period and may linger for years after the termination of the event [4]. A combination or sequence of droughts has severe impacts on human and environmental welfare [5]. Drought can be grouped into five kinds, namely, Meteorological drought; Hydrological drought; Agricultural drought, Socio-economic drought, and Groundwater drought [6]. Meteorological drought is the consequence of a natural reduction in the amount of precipitation received over an extended period, usually a season or more in length [4]. The World Meteorological Organization (WMO) has recommended using the SPI for extensive use by all national Meteorological and Hydrological Services to ascertain Meteorological drought and complement local drought indices currently being used [7]. The Southern Oscillation Index (SOI) measures a large-scale fluctuation in sea-level pressure between La Niña and El Niño. The years of abnormally high sea surface temperature (SST) from the west coast of South America towards the equatorial mid-pacific are known as El Niño years, and the years of abnormally colder waters in the same region are known as La Nina years [8]. Most drought years are associated with El Niño episodes, followed by non-El Niño events in India [9–12]. During the El Niño years, a strong tendency for below-normal Indian monsoon rainfall spread over most parts of the country [8,12,13]. There is good agreement between the significant negative/positive value

of SOI and droughts/floods; however, some exceptions are unexplained by the SOI [13]. The decrease in Indian monsoon rainfall was associated with the warm phase of ENSO due to anomalous regional Hadley circulation with decreasing motion over the Indian subcontinent [12].

Nepal observes excess rainfall in the monsoon months of a year, while there is a deficit in some other months. Almost 80% of the annual rainfall occurs during the monsoon months of June to September, and the remaining months face a deficit of water [14,15]. Summertime is dominated by a monsoonal climate, while wintertime by western disturbances. There are four seasons in Nepal, namely, pre-monsoon (March–May), monsoon (June–September), post-monsoon (October–November), and winter (December–February). Pre-monsoon is characterized by hot, dry, and westerly windy weather with mostly localized precipitation in a narrow band, whereas the monsoon is characterized by moist southeasterly monsoonal winds coming from the Bay of Bengal and occasionally from the Arabian Sea with widespread precipitation. Post-monsoon refers to a dry season with sunny days featuring the driest month, November. Winter is a cold season with precipitation mostly in the form of snow in high-altitude mountainous regions.

Many studies have been undertaken to detect rainfall variability in Nepal; however, very few studies have been conducted on drought monitoring. Although few historical drought monitoring research has been conducted, real-time drought monitoring by the national meteorological and hydrological service (NMHS) has not been carried out in Nepal. Instead, with the rainfall data, NMHS issues the seasonal precipitation analysis (above normal/normal/below normal) based on the meteorological observatories. Sigdel and Ikeda [16] analyzed the spatial and temporal variation to investigate drought patterns using the 26 stations over Nepal during 1971–2003. Similarly, Wang et al. [17] studied drought in the western region of Nepal. Likewise, Dahal et al. [18] concentrated their study on the central region of Nepal. Shrestha et al. [19] focused on the Koshi basin located in the eastern region of Nepal, while Kafle [20] focused on drought studies in Nepal's far and mid-western regions. However, drought studies using many stations and long-term data sets are still lacking.

The research hypothesis behind this study is that drought frequency and intensity is continuously increasing over Nepal in recent decades, and this could be accelerated due to large-scale atmospheric circulations, such as El Niño and Southern Oscillation. The main objective of the study was to monitor the historical drought over Nepal and its relationship with El Niño and Southern Oscillation. The specific objectives are to: (i) identify the extreme drought years during the last four decades; and (ii) study the temporal and spatial progression of drought events during El Niño and non-El Niño years.

## 2. Materials and Methods

### 2.1. Study Area

Nepal is a landlocked country located between India and China and in the southern part of the central Himalayans. It extends from 80°04′ to 88°12′ E longitude and 26°22′ to 30°27′ N latitude. It comprises 77 districts. The complex topography of Nepal ranges from the low land of Terai 60 m in the southern parts of the country, covering the fertile lands, and the high land of Mount Everest 8848 m above sea level in the Himalayan region. The northern part of Nepal has the most prominent peak worldwide. The country extends 885 km from the east to the west and varies from 130 km to 260 km in the north to the south, respectively, and covers 147,181 sq. km. Precipitation data from January 1977 to December 2018 were acquired from the Department of Hydrology and Meteorology, Government of Nepal. The spatial distributions of the 107 meteorological stations across the different regions of the country are shown in Figure 1. The amount of annual precipitation generally decreases from east to west. However, there are certain pockets with heavy annual rainfall totals, such as in central Nepal [16]. The stations measure rainfall on a daily basis and later convert it to monthly total rainfall in each station, which was applied in the present study. The stations were selected based on less than 10% of missing records,

and most of the stations (95%) were lower than 3% of the total number of annual values. High-altitude stations were used for spatial coverage with 30 years' time series with 5–10% missing values.

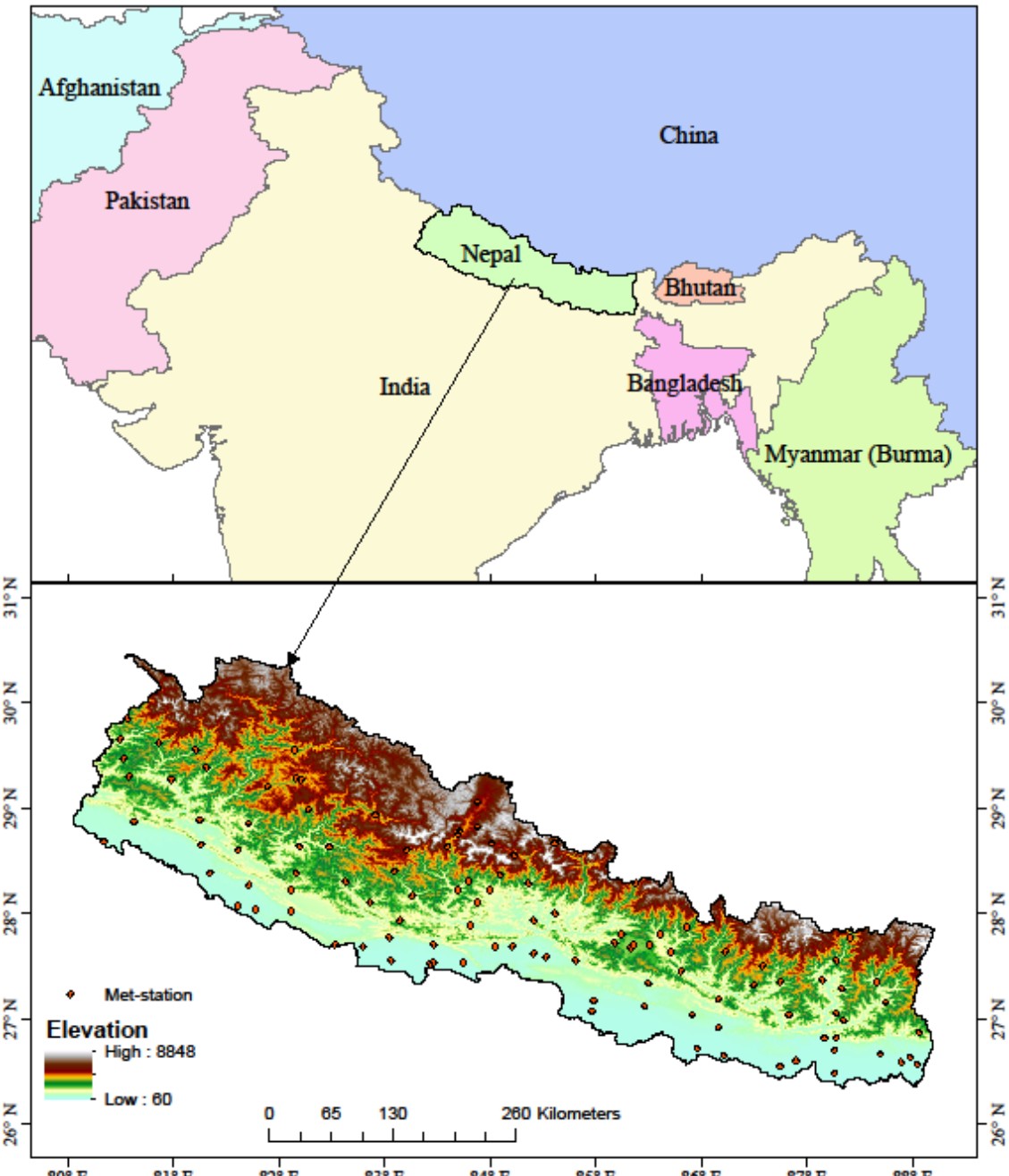

**Figure 1.** Location map of the study area along with rainfall stations at different elevations.

We adopted the Normal ratio method to estimate the missing rainfall values of the climate data set from three nearby weather stations [21]. Furthermore, the data selection criterion was based on the length, completeness, and reliability of the time series records of the stations as much as possible.

Monthly SOI data (1977–2018) were acquired from the climate Analysis Center of the National Oceanic and Atmospheric Administration (NOAA) available on https://www.cpc.ncep.noaa.gov/data/indices, accessed on 1 February 2020. In addition, descriptions of Changes to Ocean Niño Index (ONI) monthly SST anomalies over the Niño 3.4 region were obtained from the National Weather Service Climate Prediction Center of the NOAA,

https://origin.cpc.ncep.noaa.gov/products/precip/CWlink/MJO/enso.shtml, accessed on 1 February 2020 for the period 1977 to 2018.

*2.2. Standard Precipitation Index (SPI)*

SPI was developed to detect, calibrate, quantify, and monitor drought using long-term precipitation data sets [22]. SPI is normalized so that wetter and drier climates of different time scales can be represented simultaneously [23]. The duration of weeks or months and seasons can be used to apply this index to agricultural interests, and a longer duration of years can be used to apply this index to water supply and water management interests [24,25]. It has also been demonstrated in several studies in different regions of the world.

The SPI calculation method is as follows; the monthly precipitation is fitted to gamma distribution. The probability density function of Gamma distribution is defined as

$$g(x) = \frac{1}{\beta^{\alpha}\Gamma(\alpha)} x^{\alpha-1} e^{-x/\beta} \tag{1}$$

for $x > 0$, where

$\alpha > 0$ $\alpha$ is a shape parameter;
$\beta > 0$ $\beta$ is a scale parameter; and
$x > 0$ $x$ is the precipitation amount.

$$\Gamma(a) = \int_0^{\infty} y^{\alpha-1} e^{-y} dy \tag{2}$$

$\Gamma(a)$ is the gamma function.

Fitting the distribution to the data requires alpha and beta to be estimated. The maximum likelihood function is given as follows:

$$\hat{\alpha} = \frac{1}{4A}\left(1 + \sqrt{1 + \frac{4A}{3}}\right)$$
$$\hat{\beta} = \frac{\bar{x}}{\hat{\alpha}} \tag{3}$$

$$\text{where } A = \ln(\bar{x}) - \frac{\sum \ln(x)}{n} \tag{4}$$

$n$ = number of precipitation observations.

The resulting parameters are then used to find the cumulative probability of an observed precipitation event for the given month and time scale for the station in concern. The cumulative probability is given by:

$$G(x) = \int_0^x g(x) dx = \frac{1}{\hat{\beta\alpha}\Gamma\left(\hat{\alpha}\right)} \int_0^x x^{\hat{\alpha}-1} e^{-x/\hat{\beta}} dx \tag{5}$$

letting the equation $t = x/\hat{\beta}$ become

$$G(x) = \frac{1}{\Gamma\left(\hat{\alpha}\right)} \int_0^x t^{\hat{\alpha}-1} e^{-t} dt \tag{6}$$

since the gamma function is undefined for $x = 0$, and the precipitation distribution may be zero, so that the cumulative probability becomes:

$$H(x) = q + (1-q)G(x) \tag{7}$$

where $q$ is the probability of zero. $H(x)$ is then transformed to a normal distribution with a zero mean and unit variance.

The solution of the above approach is as follows:

$$Z = SPI = -\left(t - \frac{c_0 + c_1 t + c_2 t^2}{1 + d_1 t + d_2 t^2 + d_3 t^3}\right) \tag{8}$$

for $0 < H(x) < 0.5$

$$Z = SPI = +\left(t - \frac{c_0 + c_1 t + c_2 t^2}{1 + d_1 t + d_2 t^2 + d_3 t^3}\right) \tag{9}$$

for $0.5 < H(x) < 1.0$

$$\text{where, } t = \sqrt{\ln\left(\frac{1}{(H(x))^2}\right)} \tag{10}$$

for $0 < H(x) < 0.5$

$$t = \sqrt{\ln\left(\frac{1}{(1.0 - H(x))^2}\right)} \tag{11}$$

for $0.5 < H(x) < 1.0$

$c_0 = 2.515517$; $c_1 = 0.802853$; $c_2 = 0.010328$

$d_1 = 1.432788$; $d_2 = 0.189269$; $d_3 = 0.001308$

In this study, the SPI was computed at 4- and 12-month timescales using the "SPEI" package in R-statistical software. The index computation is simple and based on precipitation only as an input, and the outputs have multiple time scales of SPI. We defined a 4 month time-scale of SPI as SPI4 and a 12-month time-scale of SPI as SPI12 for summer and annual drought identification, respectively.

The threshold for indicating the severity of meteorological drought based on SPI was adopted [22]. First, the SPI4 and SPI12 data sets were interpolated using the inverse distance weighted (IDW) function [24]. Then, the interpolated maps were reclassified into different severity classes using the SPI threshold (Table 1). For example, the interpolated SPI4 of the months of September was reclassified because the SPI4 of September comprises the accumulated precipitation total of the rainfall received in June, July, August, and September, which is crucial to the significant cropping season in Nepal. Similarly, the interpolated SPI12 of the December months was reclassified because the SPI12 of December comprises the accumulated precipitation total of the rainfall from January to December, which is crucial for water resource planning.

**Table 1.** SPI classification thresholds based on the study by McKee et al. [22].

| SPI Values | Drought Category |
| --- | --- |
| 0 to –0.99 | Mild drought |
| –1.00 to –1.49 | Moderate drought |
| –1.50 to –1.99 | Severe drought |
| –2.00 or less | Extreme drought |

Time series of SPI for individual stations were obtained to determine the variation in SPI time length. Time series were also utilized to observe the variation in detecting the drought length of data sets. A typical drought year was detected from the average SPI time series analysis.

The nonparametric rank-based Mann-Kendall test (M.K.) was applied to assess the time series data [26,27]. Many researchers have used this method [25,28,29] to detect trends in climatic time series data in different regions of the globe.

## 3. Results

### 3.1. Trend Analysis of SPI4 and SPI12

The results of the individual station trend analysis of SPI4 and SPI12 were examined in different regions of Nepal, which are summarized in Figure 2a,b, and the SPI4 and SPI12 trends were identified using the MAKESENS Template [30], and the MAKESENS program was developed for detecting and estimating trends in the time series of annual values of atmospheric and precipitation concentrations by the Finnish Meteorological Institute depending on the Z value (positive, negative, and zero) of individual stations, which are shown in Figure 2a,b. The Mann-Kendall trend test identified increasing, decreasing, and no trends of SPI4 and SPI12 in different regions. Downward red triangles and upward blue triangles represent negative and positive trends, respectively. In contrast, small downward red and upward blue triangles represent negative and positive trends, and big triangles indicate significant trends. However, green circles indicate no trends.

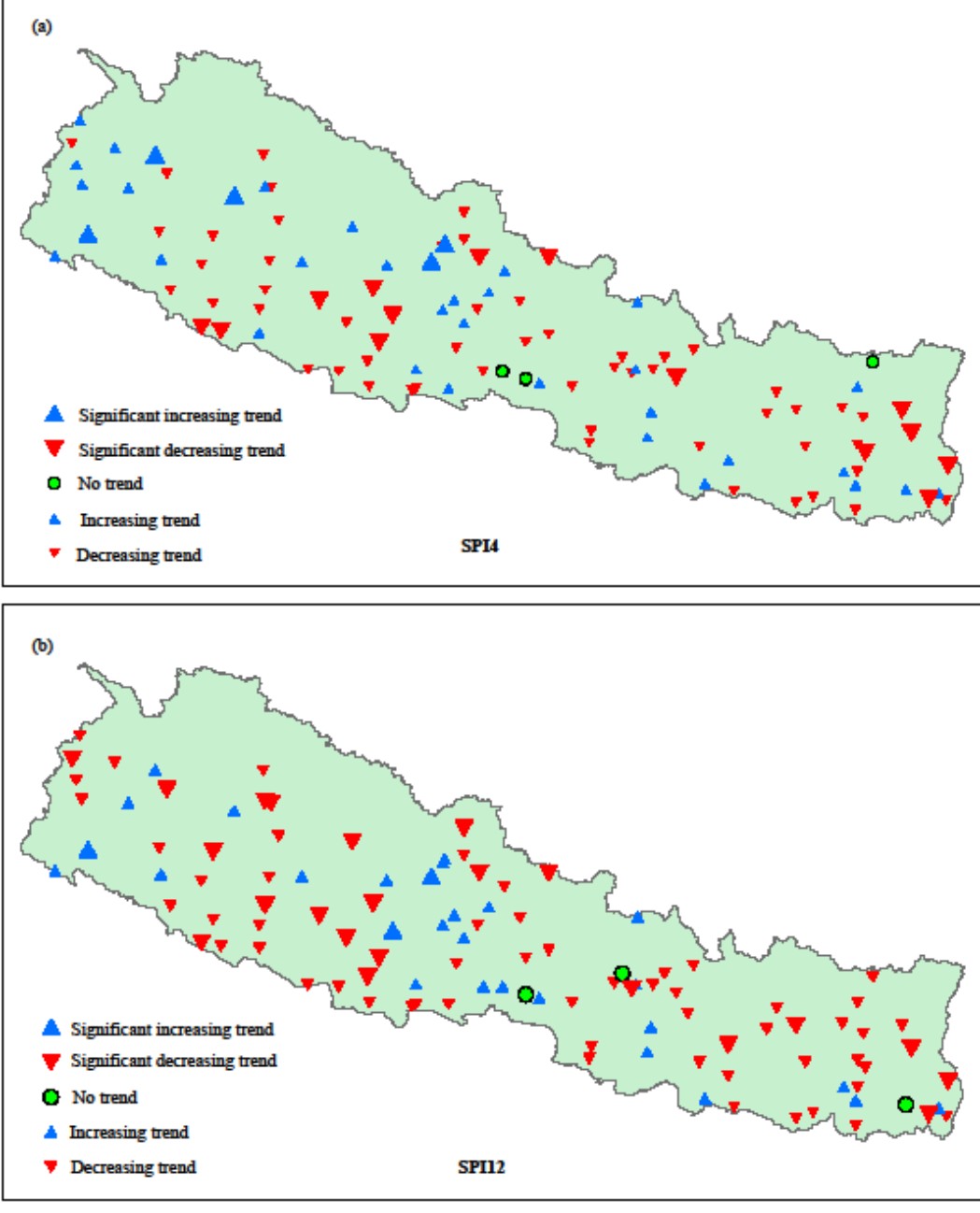

**Figure 2.** Station-wise (**a**) summer and (**b**) annual trends over Nepal.

For SPI4 among the 107 stations, 14 stations showed a significant decreasing trend, and five stations showed a significant increasing trend. Among the 14 significant decreasing trends, 3 were recorded in western, 5 in central, and 6 in eastern Nepal.

Similarly, for SPI12, 22 stations showed a significant decreasing trend, and three stations showed a significant increasing trend. Among the 22 significant decreasing trends, 8 were recorded in the western region, 8 in the central region, and 6 in the eastern region of Nepal, respectively. In summary, for SPI4 and SPI12, negative trends seem to be more common for Nepal's eastern region than the central and western regions of Nepal.

### 3.2. Temporal Variability of SPI4 and SPI12

We applied stations' average monthly precipitations to generate the temporal pattern for SPI4 and SPI12, which is depicted in Figure 3a,b. Summer drought seasons were identified based on the SPI4 intensity. A season is defined as a drought season when the SPI thresholds are $<-1$. The seasons are categories for the severity of the drought that depend on threshold values. They are shown in Table 1. Among 42 summer seasons, the eight summer drought years are 1977, 1979, 1982, 1992, 2005, 2006, 2009, and 2015 (Figure 3a). The SPI4 intensity is below 0 in 19 cases, and above for 23 cases. The magnitude of SPI4 was about $-1.96$ in 1992, and followed in 2015 at about $-1.85$.

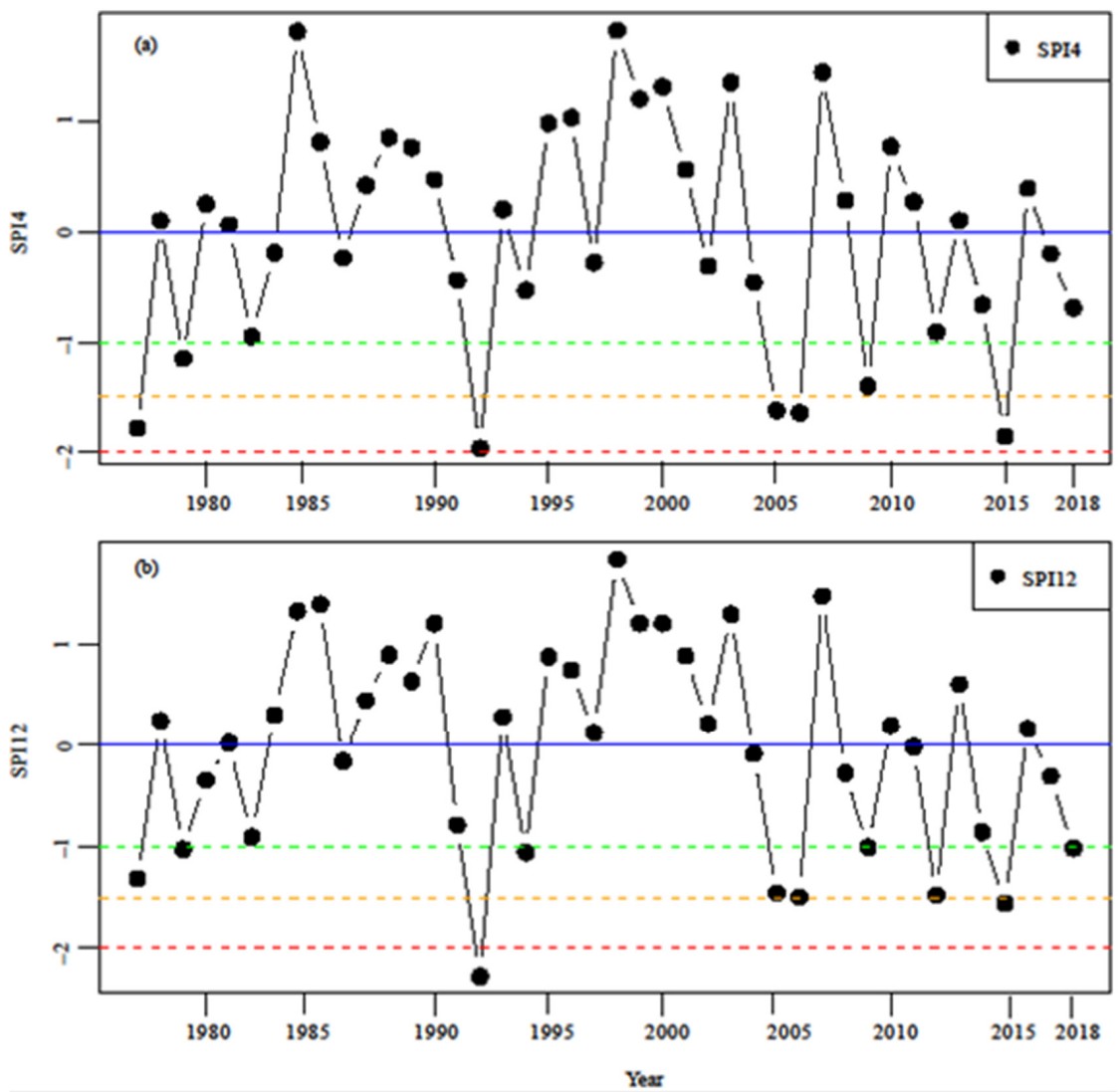

**Figure 3.** Temporal variability of SPI for (**a**) summer (SPI4), (**b**) annual (SPI12).

Similarly, the annual drought years were identified based on the SPI12 intensity. Based on the threshold value of Table 1, the 10 identified drought years were (1977, 1979, 1992, 1994, 2005, 2006, 2009, 2012, 2015, and 2018) (Figure 3b). The precipitation deficits were observed for 19 cases, and in excess for 23 cases. The magnitude of SPI12 ranges from −2.28 in 1992 and 1.83 in 1998. Analyzing both time-scales of SPI4 and SPI12, we have identified that most of the drought years coincide between the time-scales. The region behind this can be the weightage of summer precipitation which the annual one follows.

### 3.3. Spatial Distribution of Worst Drought Events

Based on the drought magnitude for both SPI4 and SPI12, 8 summer drought years and 10 annual drought years were identified (Tables 2 and 3). The two most widespread summer drought years were 1992 and 2015, where larger areas were affected by drought severities. In those years, the central region of Nepal was affected more than the eastern and western regions of Nepal.

**Table 2.** Summer drought severities based on stations' proportions expressed in percentages in different years.

| Rank | SPI4 Values | Year | Extreme | Severe | Moderate | Mild |
|------|-------------|------|---------|--------|----------|------|
| 1 | −2.0 | 1992 | 7.5 | 11.2 | 25.2 | 41.1 |
| 2 | −1.9 | 2015 | 9.7 | 10.7 | 25.2 | 38.8 |
| 3 | −1.8 | 1977 | 11.3 | 10.3 | 18.6 | 37.1 |
| 4 | −1.6 | 2006 | 7.5 | 10.3 | 15.9 | 42.1 |
| 5 | −1.6 | 2005 | 4.7 | 8.4 | 15.9 | 48.6 |
| 6 | −1.4 | 2009 | 6.5 | 3.7 | 15.0 | 57.0 |
| 7 | −1.2 | 1979 | 14.4 | 6.7 | 7.7 | 39.4 |
| 8 | −1.0 | 1982 | 3.8 | 2.9 | 16.2 | 44.8 |

**Table 3.** Annual drought severities based on stations' proportions expressed in percentages in different years.

| Rank | SPI12 Values | Year | Extreme | Severe | Moderate | Mild |
|------|--------------|------|---------|--------|----------|------|
| 1 | −2.3 | 1992 | 8.4 | 22.4 | 24.3 | 36.4 |
| 2 | −1.6 | 2015 | 6.8 | 8.7 | 23.3 | 37.9 |
| 3 | −1.5 | 2006 | 6.5 | 9.3 | 12.1 | 51.6 |
| 4 | −1.5 | 2012 | 8.5 | 11.3 | 17.9 | 40.6 |
| 5 | −1.5 | 2005 | 4.7 | 8.4 | 15.9 | 40.2 |
| 6 | −1.3 | 1977 | 8.2 | 10.3 | 11.3 | 36.1 |
| 7 | −1.1 | 1994 | 2.8 | 10.3 | 16.8 | 41.1 |
| 8 | −1.03 | 1979 | 7.7 | 4.8 | 16.3 | 36.5 |
| 9 | −1.02 | 2018 | 5.1 | 12.2 | 11.2 | 41.8 |
| 10 | −1.01 | 2009 | 4.7 | 2.8 | 16.8 | 46.7 |

For SPI4 in the year 1992, no drought condition was observed over some regions of northwestern Nepal and for a few areas of the eastern mountain. For this year, moderate drought dominated from the central to eastern–southern parts, while during 2015, moderate drought dominated over mid-west and central Nepal. Approximately 8 and 10 percent of the stations were affected by extreme drought conditions in 1992 and 2015. About 11 percent of stations were affected by severe, and around 25 percent of the stations were affected by moderate drought conditions for SPI4 in the years 1992 and 2015 (Figure 4a,c). More than 76 percent of stations had negative SPI4 values; thus, large parts of Nepal faced the precipitation deficit in both worst years.

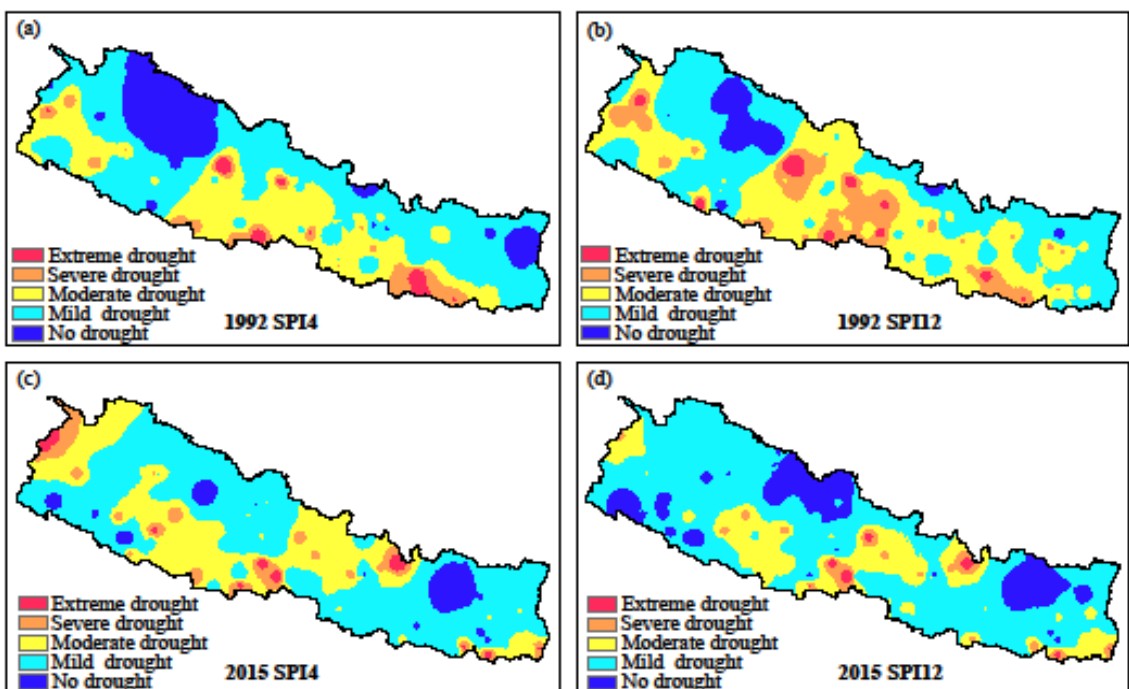

**Figure 4.** Spatial distributions of (**a**) SPI4 of 1992, (**b**) SPI12 of 1992, (**c**) SPI4 of 2015, and (**d**) SPI12 of 2015.

Similarly, for SPI12 during the drought years of 1992 and 2015, larger areas of the central region were affected by dry conditions. In the year 1992, the central region was highly affected by the dryness than the 2015. Almost the entire area was dominated by drought conditions, except a certain region of northwestern Nepal in 1992. About 93 and 78 percent of stations recorded negative values of SPI12 in 1992 and 2015, respectively (Figure 4b,d). Therefore, large areas of Nepal faced a precipitation deficit.

*3.4. Spatial Distribution of Other Major Summer Drought Events*

The spatial variation of severities of summer drought events was uniquely different among the different drought categories. The severities of drought are clearly shown in Figure 5a–f. The mean SPI4 intensity of the summer drought years ranked from 3rd to 8th ranges from −1.78, to −1.

In the year 1977, a large area in the lower belts of the eastern region and mid-mountain areas were more affected by the summer drought than the central and the western regions of Nepal (Figure 5a). In the year 2006, the central region was affected more by the severe and moderate drought conditions than any of the other regions. Similarly for the year 2005, the central region of Nepal was highly affected by extreme drought and followed in the upper parts of the mountains of western Nepal; however, low lands of Nepal were rarely affected. In 2009, drought shifted to the lower regions of Nepal. However, the central high mountain and the mid belt of the eastern region were affected. In 1979, the only western region of Nepal was affected by dryness; however, the eastern regions of Nepal had no drought situation. During 1982, the eastern regions were mainly affected by extreme, severe, and moderate drought conditions, but the central and western regions of Nepal were in normal conditions. Drought events during different years for SPI4 with the percent coverage are tabulated in Table 2.

*3.5. Spatial Distribution of Other Major Annual Drought Events*

For SPI12, the 3rd to 10th ranks of drought mean intensity varies from −1.5 to −1.01 (Table 3). The spatial variation of drought severities is shown in Figure 6a–h. In the drought year 2006, drought was randomly affected in Nepal. However, the lower belts of Nepal were more affected by drought severities than high lands. The eastern region recorded the

drought in larger areas than the central and western regions of Nepal in the drought year 2012. About 30 percent of the country was affected by drought severities in 2005 and 1977. The western region of Nepal was highly affected by extreme, severe, and moderate drought than the central and eastern regions of Nepal in the year 2005. In 1994, the central region of Nepal had normal conditions, but the low land of the eastern and the western regions of Nepal were highly affected. For the year 2018, the eastern region of Nepal was more affected, following the central region of Nepal. However, the western region of Nepal had a usual condition. In 2009, small areas of the lower belt of Nepal were affected by higher drought severities instead of the large areas of Nepal that were affected by mild drought. There was no drought in the far western regions of Nepal in the year 2009. The details of drought categories during different years for SPI12 with the percent coverage are tabulated in Table 3.

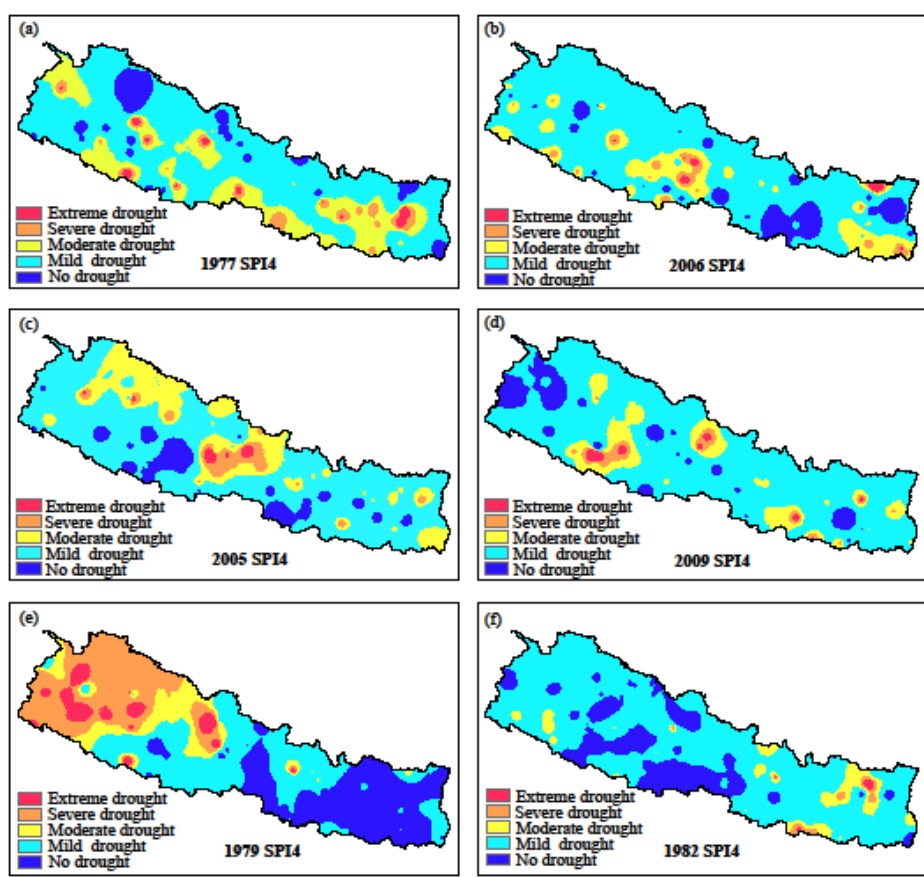

**Figure 5.** Spatial distributions of (**a**) summer (SPI4) of 1977, (**b**) summer (SPI4) of 2006, (**c**) summer (SPI4) of 2005, (**d**) summer (SPI4) of 2009, (**e**) summer (SPI4) of 1979, and (**f**) summer (SPI4) of 1982.

### 3.6. Relationship between SPI and Climate Indices

This study used the SOI and ONI index to show the relationship with summer and annual SPI. The SOI and ONI index measure large-scale fluctuations in sea level pressure and temperature over region 3.4. Similar studies have been conducted around the globe [31].

The comparison between the SPI4 and the summer SOI series shows that there were 32 events (76.17%) phase relation (positive/negative SOI, positive/negative SPI4), and 10 (23.81%) events show the in-phase relationship in Figure 7a. The correlation coefficient between SPI4 and summer SOI was 0.52 at a 95 percent confidence level. Thus, SPI4 and the summer SOI series are highly correlated. Generally, during the drought/flood period, SOI and the SPI are (−/+), which is stronger than the usual years. Generally, during the drought years (El Niño and non- El Niño years), low summer SOI and negative summer SPI in Nepal are in phase relations on warm phase periods. Deficit/excess conditions

from the above-mentioned relationship between SPI4 and summer SOI results; the SPI4 is influenced from the summer SOI.

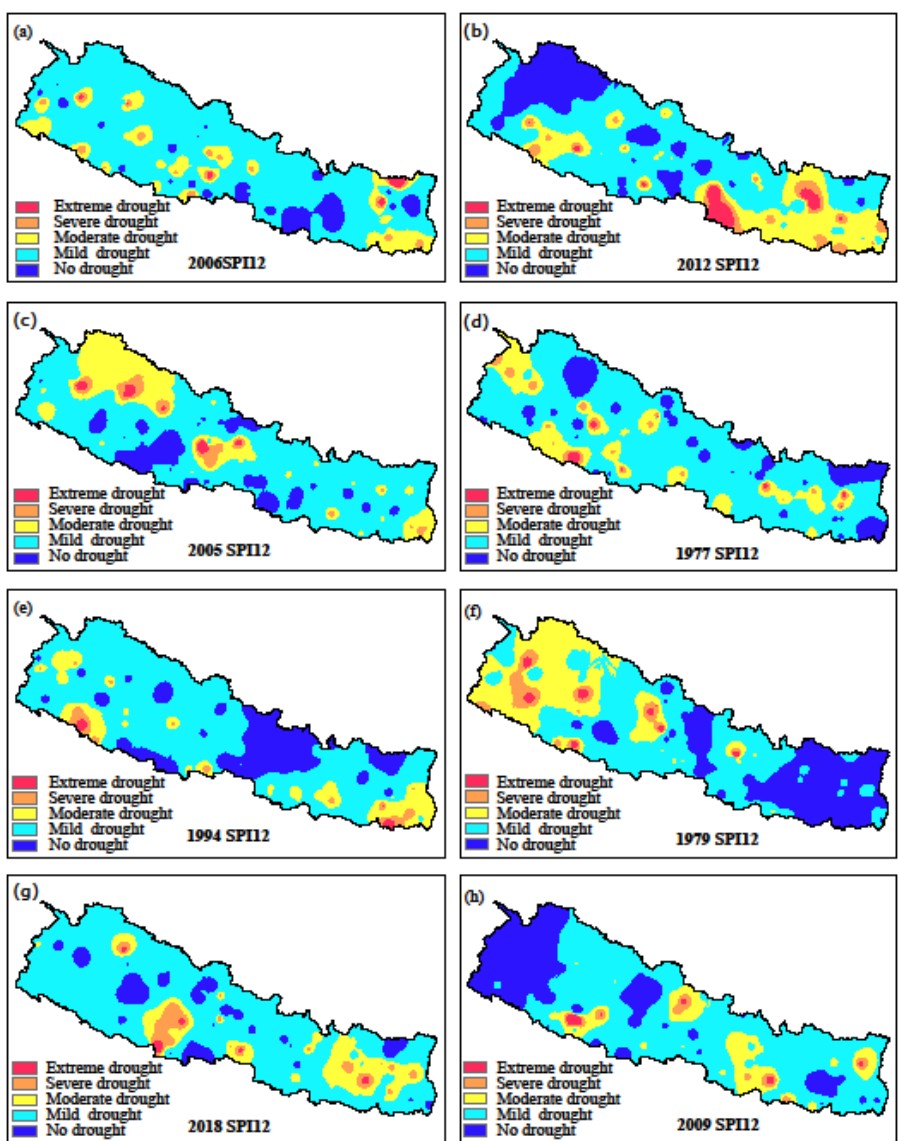

**Figure 6.** Spatial distributions of (**a**) annual (SPI12) of 2006, (**b**) annual (SPI12) of 2012, (**c**) annual (SPI12) of 2005, (**d**) annual (SPI12) of 1977, (**e**) annual (SPI12) of 1994, (**f**) annual (SPI12) of 1979, (**g**) annual (SPI12) of 2018, and (**h**) annual (SPI12) of 2009.

Similarly, a comparison between SPI12 and annual SOI analysis indicates that there were 21 events (about 50%) and the phase relation (positive/negative annual SOI, positive/negative SPI12) is clearly shown in Figure 7b. Among the 21 events, 10 events (47.62%) were in deficit and 11 events (52.38%) were in excess. The correlation between SPI12 and annual SOI is 0.14 at a 95% confidence level. The deficit/excess condition shows that the SPI12 series is influenced from the annual SOI. However, during the drought/flood period, annual SOI and the SPI12 are stronger ($-/+$)than the usual years.

The summer drought years were also further linked with ONI (El Niño and La Niña episode) which was extracted from https://origin.cpc.ncep.noaa.gov/products/precip/CWlink/MJO/enso.shtml, accessed on 1 February 2020. The El Niño and La Niña years are tabulated in Table 4.

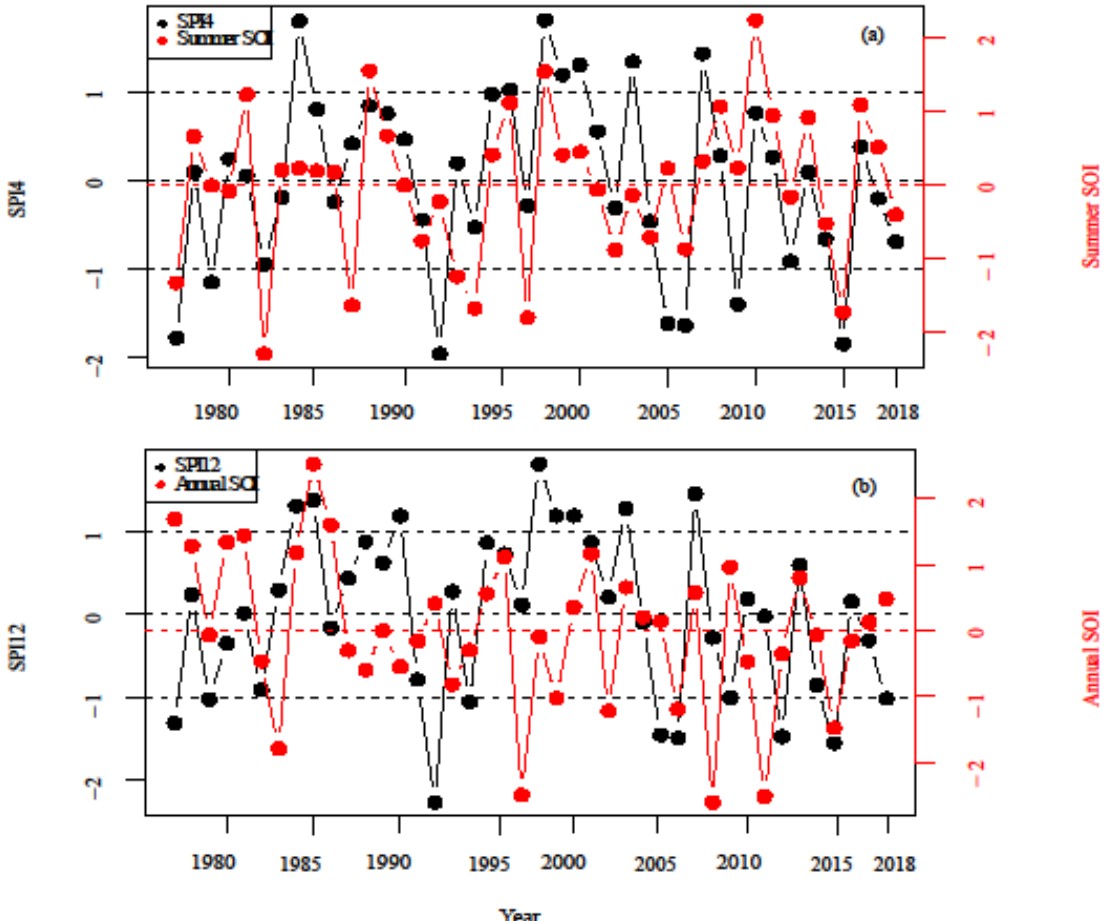

**Figure 7.** Relationship between SPI and SOI: (**a**) summer, (**b**) annual.

**Table 4.** The El Niño and La Niña years from 1977 to 2018.

| Condition | Years |
|---|---|
| El Niño | 1982, 1983, 1987, 1991, 1992, 1997, 1998, 2002, 2004, 2009, 2015, 2016 |
| La Niña | 1985, 1988, 1989, 1995, 1998, 1999, 2000, 2007, 2008, 2010, 2011, 2016 |

There were eight excess rainfall years in Nepal that coincided with La Niña years. Out of eight summer drought years in Nepal, four years coincided with El Niño years and the other four drought years were marked in non-El Niño years.

## 4. Discussions

The trend analysis of SPI4 and SPI12 showed a decreasing trend over most of the stations, which indicates the increasing drought frequency. The negative trends are steady for both time-scales among a majority of the stations, which was caused because monsoon (4 months JJAS) rainfall dominated the annual. The results are consistent with the Ref. [18].

Our results depict the fact that the frequency and intensity of drought events increased in recent years. For SPI12, we have identified 10 major years, while for SPI4, 8 major drought years were identified. The result is similar to the findings presented by previous researchers [15] during the summer drought years of 1977, 1982, 1991, and 1992. Furthermore, the study by the Refs. [15,32] indicates the deficit years based on precipitation were 1957, 1972, 1977, 1982, and 1992 in Nepal.

Drought severities depend on the intensity of SPI4 and SPI12. Eight summer droughts and ten annual drought events have been revealed, and each drought event has unique SPI dynamics. There has been no similar drought pattern over Nepal in drought events

over the past four decades. The spatial variation of SPI4 and SPI12 indicates that every drought episode has different severities (extreme, severe, moderate, and mild drought) with differences in SPI intensities.

The areas of Nepal affected by the average extreme, severe and moderate summer drought were 8, 9, and 18 percent on summer drought events over the study period. Similarly, the areas of Nepal affected by the average extreme, severe and moderate annual drought were 7, 11, and 17 percent on annual drought events. Previous researchers [16] noticed that the average extreme, severe, and moderate annual droughts were 5, 5, and 9 percent concerning SPI12 over the period (1971–2003). A slight difference can be observed in between the severities in this study. The reason behind this is that in recent years, the worst drought events happened more frequently. On average, mild drought covered 42 percent in summer and 40 percent in annual drought events. On average, Nepal's dry areas were found to be 77 percent in summer events and 75 percent in annual drought events.

The monsoon rainfall variability of Nepal is influenced by local effect, the regional circulation system, and large-scale circulation from the Indian and Pacific Ocean. Strengthening/weakening of the rainfall pattern has been identified and correlated with La Niña/El Niño years. The analysis revealed a relationship between the SPI4 and SPI12 with large-scale climate indices. Among them, SPI4 is well-correlated with SOI above a 95 percent confidence level. Previous researchers were concerned with the relationship between SPI3 and SPI12 time-scales with summer and annual SOI [16]. Similarly, the Refs. [15,32] focused on the relationship between summer rainfall of Nepal and SOI. The extensive circulation system affects the monsoon rainfall during the La Niña and El Niño episodes more than the average years. The drought events on El Niño years and non-El Niño years were strongly related between SPI and SOI than the average years. The relationship between SPI and the climate indices such as the SOI and ONI anomaly over the Niño 3.4 has suggested that one of the causes for summer droughts is El Niño.

This study indicated that summer droughts occurred in both El Niño and non-El Niño years. Out of eight drought years, only four drought years were associated with El Niño episodes (1982,1992, 2009, and 2015), and the remaining four drought years (1977,1979, 2005, and 2006) were recorded in non-El Niño years. Similarly, annual droughts evolved in El Niño and non-El Niño years. Out of 10 drought years, only 3 drought years were associated with El Niño years (1992, 2009, and 2015) and 7 drought years (1977,1979, 1994, 2005, 2006, 2012, and 2018) were recorded in non-El Niño years. Each drought event also had distinct SPI dynamics.

South Asian countries have recently been experiencing frequent drought incidents, and the Standardized Precipitation Index (SPI) has mostly been adopted in South Asian countries to quantify and monitor droughts [33]. Khatiwada and Pandey [34] identified that the SPI is focused foremost on capturing the duration and intensity of drought in the Karnali river basin, Nepal. Using SPI from 1951–2015 over India, the analysis revealed that of the 34 met sub-divisions, 16 sub-divisions during the annual and monsoon season were drought-prone and showing high negative SPI values [35]. Similarly, the occurrence of drought with negative SPI values is frequent in West Bengal districts in India with increasing dry events and decreasing wet and normal events [36]. The present study also showed similar patterns of SPI as aforementioned studies. El Niño alone is not to be blamed for droughts in India. A total of 10 out of 23 droughts that India witnessed have occurred during years when El Niño was absent [37]. The current study is consistent with their findings too.

Nepal has experienced consecutive and worsening drought conditions in a severe drought episode during 2005–2006. These events were one of the most challenging events for the mountainous country Nepal for agriculture practices and water resource management. Since drought directly affects crops leading to economic losses, countries like Nepal, where people are primarily dependent on rain-fed agriculture for their livelihoods, are very vulnerable due to drought [38]. The livelihoods of around 60 percent of the total Nepalese

population are directly dependent on agriculture [39]. Extreme droughts negatively impact both cash and cereal crops [40]. Therefore, impact assessment studies of drought events are essential. These outputs help make decisions for water resource management and water allocations for mitigating the impacts of drought.

## 5. Conclusions

This study provided concise knowledge about the temporal and spatial variability of meteorological droughts using a SPI4 time-scale and SPI12 time-scale over Nepal during the past four decades (1977–2018). Among the drought events, 1992 was the worst summer and annual drought year, followed by 2015. Extreme drought events of 1992 and 2015 indicated more drought signals in the central region than any other regions of Nepal. Most of the study sites show a negative trend for SPI4 and SPI12, which indicates the drought episodes are increasing and being more frequent.

In the last four decades, the drought events in Nepal indicated extreme, severe, moderate, and mild categories with reference to the intensity. The obtained proportional weightages of summer drought severities for extreme, severe, and moderate drought were 8, 9, and 18 percent, respectively. Similarly, the obtained proportional weightages of annual drought severities for extreme, severe, and moderate drought were 7, 11, and 17 percent, respectively. All drought categories were more highly affected in the western and central regions than the eastern region of Nepal, with some exceptions.

The relationship of SOI with SPI is strong in summer (SPI4) and weak in annual (SPI12). This study revealed that summer droughts were recorded both in El Niño and non-El Niño years. During eight summer drought years, only four drought years were associated with El Niño episodes.

The study's findings can help make decisions for water resource management and water allocations for mitigating the impact of droughts. It is also beneficial to the researchers to study climate change in Nepal and the sustainable development of regional water resources.

**Author Contributions:** Conceptualization, D.B., D.A. and M.S.; Data analysis and original draft preparation, D.B.; Review and editing, M.S.; Supervision, D.A.All authors have read and agreed to the published version of the manuscript.

**Funding:** This research received no external funding.

**Institutional Review Board Statement:** Not applicable.

**Informed Consent Statement:** Not applicable.

**Data Availability Statement:** Not applicable.

**Acknowledgments:** Nepal Government's Department of Hydrology and Meteorology is acknowledged for providing observed rain gauge data, and University Grant Commission, Kathmandu Nepal for financial support (U.G.C. Award No: PhD-75/76-S&T-11) was provided to the first author. Climate index data were taken from the National Oceanic and Atmospheric Administration (NOAA).

**Conflicts of Interest:** The authors declare no conflict of interest.

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
