# Peer review of "Drought Monitoring over Nepal for the Last Four Decades and Its Connection with Southern Oscillation Index"

_water, doi:10.3390/w13233411_

Round 1

Reviewer 1 Report

The purpose of this paper was to capture drought Events over Nepal and also to track its linkage with the Southern Oscillation Index (SOI) by employing the Standardized Precipitation Index (SPI) and the Pearson r correlation coefficient.  The overall design is good; it is an interesting study and well written study with good readability and results which in general presents a specific, easily identifiable advance in knowledge regarding drought the linkage between SOI and drought. Thus, I would recommend this manuscript to be accepted for publication.

Author Response

Thank you for accepting our manuscript.

Sincerely

Damodar Bagale

Reviewer 2 Report

This article looks at the drought in Nepal. The topic is timely and significant due to climate change. Overall it is well written, however, I have a few comments to help improve the paper:

The article lacks a specified research objective and research hypotheses. Please add this information. Otherwise, the manuscript will not qualify as a research article.

What is missing in the introduction is information on how drought monitoring and meteorological monitoring are done in Nepal. Please elaborate on this thread especially since you address this topic in the title of the paper.

Try re-writing the abstract using the impersonal form rather than we. It will get a more scientific tone.

The discussion section should be developed further. At present, it seems to be short. Please expand it concerning the world literature. Has anyone else studied similar relationships? What is the pattern of drought in Nepal's neighbouring countries? Are SPI calculations sufficient for determining the extent of drought? Or do global studies show otherwise?

Editorial Notes:

In figure 1, please increase the size of the captions - they are currently unreadable.

In the description of Table 1, the citation - McKee et al., 1993 should also be written as a number in brackets (like other citations)

The list of references is not adapted to the requirements of the journal. Please check the reference list guidelines and improve the list. Please note which words should be in bold and which in italics.

Author Response

Please find the attached point to point reply report to reviewer.

This manuscript is a resubmission of an earlier submission. The following is a list of the peer review reports and author responses from that submission.

Round 1

Reviewer 1 Report

This is a neat little study of drought in Nepal. However, I do find multiple areas that needs improvement, and cannot recommend the article for publication in its current form.

General comments:

  1. the word percentages is incorrectly used throughout the manuscript. Change it to percent, or simply %.
  2. It would be interesting with a bit more background on the implications of drought in Nepal. The journal has an international audience, many of who aren’t very familiar with the hydroclimatology and water resources of Nepal.
  3. The result section is too detailed on reporting the exact % of drought in each year and category. This is unnecessary, as the reader can see this in tables 2-3 and the interpolated maps, so this section can be shortened. Instead it would be interesting to see an explanation of the spatial patterns of drought.
  4. The connection to the SOI index needs further explanation. It seems like SPI4 might be positively correlated with SOI, but this is poorly described in the text, instead the reader gets the impression that both ENSO extremes cause drought in Nepal, which isn’t the case.
  5. Analysis including the ONI index is lacking from the results.
  6. The whole section about trends in SPI needs to be clarified. Consider only reporting significant results in the text.
  7. Large parts of the discussion aren’t supported by the results presented (for example the first paragraph, and the paragraph starting at line 384). The same is true for the conclusion.

Specific comments:

Rows 16-18: “The areas of Nepal affected by average extreme, severe and moderate drought both in summer and annual events are 7, 17 9,18 percentages and 7,11,17 percentages respectively.” I had to re-read this sentence a few times to understand it. Please re-write it to make it clearer.

Row 30: add “time” between “considerable” and “period”

Rows 36-38: Add a reference to this statement

Row 41: this should be change to La Niña years.

Rows 42-46: Please add a physical explanation here, how do ENSO events affect the regional circulation to cause drought in India? A definition of the SOI is also needed.

Row 76-77 add a reference for this method.

Row 78-84: add full references to the data sets in the reference list.

Row 103: Define SPI4 and SPI12.

Row 104: share what the threshold you used is.

Row 106: add reference on IDW.

Row 128: add reference to the template

Row 133: green circles

Row 150. The working is confused here, isn’t SPI4 referring to summer droughts? The wording “a year is defined as a drought year...” seem to refer to SPI12. Consider using “season” instead.

Rows 178-179: Result and discussion of temporal and spatial evolution of the droughts is missing.

Rows 184-187: Please clarify what you mean here.

Rows 198-199. Please re-write to clarify what is meant by this sentence.

Row 335-336: Put these years in a table rather than in the text. Also, add a full reference.

Rows 346-347: Avoid listing all the years like this.

Table 2. I assume this shows % of Nepal’s area that were in the different drought categories? Re-write the table caption to make this clear. Also, consider including another column of % area in mild drought, as row 192 says 70% of Nepal was in drought although that is not reflected in the tables.

Author Response

Find the attached file of responses of reviewer 1 comments

Reviewer 2 Report

The purpose of this paper was to capture drought Events over Nepal and also to track its linkage with the Southern Oscillation Index (SOI) by employing the Standardized Precipitation Index (SPI) and the Pearson r correlation coefficient.  The overall design is good; it is an interesting study and well written study with good readability and results which in general presents a specific, easily identifiable advance in knowledge regarding drought the linkage between SOI and drought. However, the following minor revisions must be taken into account in order to optimize the manuscript so as to be suitable for publication. However, the following minor revisions must be taken into account in order to optimize the manuscript so as to be suitable for publication.

  1. The title could be changed to: …. its connection with Southern Oscillation Index

  1. Line 18: Southern Oscillation Index (SOI)

  1. In the Keywords it must be added the "Mann-Kendall test".

  1. Line 58: It should be mentioned the means which were utilized (SPI index, Mann-Kendall test and others) so as to achieve this study goals.

  1. Regarding the selection of the Normal ratio method so as to fill the missing records, the following citation could be used: Myronidis, D.; Theofanous, N. Changes in climatic patterns and tourism and their concomitant effect on drinking water transfers into the Region of South Aegean, Greece. Stochastic Environmental Research and Risk Assessment. 2021, 35(9), 1725-1739.

  1. Line 120: The basic equations of the Mann-Kendall trend test must be added.

  1. Line 122: The software which was used so as to track drought and to investigate trends must be added.

  1. Line 305: It could be added that this relation was investigate by using the correlation coefficient which has been used in similar studies around the globe Myronidis, D.; Fotakis, D.; Ioannou, K.; Sgouropoulou, K. Comparison of ten notable meteorological drought indices on tracking the effect of drought on streamflow. Hydrol. Sci. J . 2018, 63:15-16, 2005-2019.

  1. Line 383: This statement could be further support by using a reference from similar studies.

  1. Line 439: Use the first letter of your first name and the first letter of your last name for all authors.

  1. References: All the references must be reformatted according to the MDPI standards.

Author Response

Please find the attached file of response to reviewer 2 comments

Reviewer 3 Report

The authors analyse a very important and crucial problem for the futurre of the area. The detection of the impact of climate change on rain, and therefore the changes on drought is crucial for a proper water management.

There are little modifications related to the article.

In line 37 they use SPI without explanation, in abstract and in line 45 appear SOI (only at line 382 you can find the explanation). In line 303 appear ONI, in line 382 you can find the real name.

I think that in matherial and methods could be useful insert a description of the climate of Nepal with some values regarding pluviometric distribution during the year in different areas. In the study area the SPI are calculated by data coming from stations with a very great difference in altitude from 72 to 3650 meter. May be there is a strong difference on the rainfall cumulated during the period of summer (SPI4) and annual (SPI12) in this different stations/areas. The impact of an increasing drougth is depending also in the amount of rainfall. In figure 2 it seems that in SPI12 there is a more evident decreasing trend respect to SPI4, is it possibile to explain this.

In line 127 the figure is number 2 not 3?

Please check the sentence in line 162 and 163, what mean "The region behind this....

In table 2 the legend could be more descriptive, I think it is more clear insert that the percentage is referred to the stations that present different value of drought severities.

In line 201 and 202, 7 is related to 1992 and 10 to 2015, please check and change.

I don't understand well the sentence on line 207 in table I can find a spatial frequency?

In the paper I don't find reference to the tables S2 where there is the SPI4 intensity.

In line 222 there is a value of -1.1.5 that is wrong

Please check the correlation between number in table 2 and 3 and the description on the text. For example in line 237 for drought in 2005 you put for extrem, severe and moderate 5, 9, 12 but in table 2 are 5, 9, 16, the same in line 249 for 1982. In line 277 and 278 concerning 2005 and 1977 and after insert 5, 9, 16 that are only for 2005, please clarify.

In line 311 and 323 the figure is 7 and not 3.

In conclusion there are some repetitions of what has already been write in discussion. I think that a broader description of the causes of this increase in drought could be useful to better understand the phenomenon and especially the different impact between the 3 areas of Nepal. You write that drought events affected more in the western and central regione than the eastern region of Nepal, why?

Author Response

Find the attached revised manuscript.

Author Response

Find the attached file of revised version manuscript.

Round 2

Reviewer 1 Report

I do not find that the edits are thorough enough to grant publication in Water. 
